# The Influence of Healthy Habits on Cognitive Functions in a Group of Hemodialysis Patients

**DOI:** 10.3390/jcm12052042

**Published:** 2023-03-04

**Authors:** Piotr Olczyk, Patryk Jerzak, Krzysztof Letachowicz, Tomasz Gołębiowski, Magdalena Krajewska, Mariusz Kusztal

**Affiliations:** Department of Nephrology and Transplantation Medicine, Wroclaw Medical University, 50-367 Wroclaw, Poland

**Keywords:** hemodialysis, cognitive functions, risk factors

## Abstract

(1) Background: Cognitive impairment (CI) is more prevalent in hemodialysis (HD) patients than in the general population. The purpose of this study was to examine if behavioral, clinical, and vascular variables are linked with CI in individuals with HD. (2) Methods: Initially, 47 individuals with chronic HD volunteered to participate in the trial, but only 27 patients ultimately completed the Montreal Cognitive Assessment (MoCA) and the Computerized Cognitive Assessment Tool (CompBased-CAT). We collected information on smoking, mental activities, physical activity (Rapid Assessment of Physical Activity, RAPA), and comorbidity. The oxygen saturation (rSO2) and pulse wave velocity (PWV; IEM Mobil-O-Graph) of the frontal lobes were measured. (3) Results: Significant associations were discovered between MoCA and rSO2 (r = 0.44, p = 0.02 and r = 0.62, *p* = 0.001, right/left, respectively), PWV (r = −0.69, *p* = 0.0001), CCI (r = 0.59, *p* = 0.001), and RAPA (r = 0.72, *p* = 0.0001). Those who actively occupied their time during dialysis and non-smokers achieved higher cognitive exam results. A multivariate regression study demonstrated that physical activity (RAPA) and PWV had separate effects on cognitive performance. (4) Conclusions: Cognitive skills are related to inter-dialysis healthy habits (physical activity, smoking) and intra-dialysis activities (tasks and mind games). Arterial stiffness, oxygenation of the frontal lobes, and CCI were linked with CI.

## 1. Introduction

The prevalence of treated end-stage kidney disease (ESKD) has increased worldwide, likely due to improving ESKD survival, population demographic shifts and increasing access to dialysis programs in countries with growing economies. The unadjusted 5-year survival of ESKD patients on kidney replacement therapy was 41% in the USA, 48% in Europe, and 60% in Japan [1]. Hemodialysis is the most common modality of kidney replacement therapy. In 2020, approximately 786,000 people in the United States had ESKD, 71% of whom were dialysed [2]. Reduced quality of life, especially in the area of mental health, is still the subject of research in this group of patients.

Mild cognitive impairment (MCI) is found in 30% to 60% of the overall population of dialysis patients, and it involves persistent cognitive impairment and behavioural disturbances. One of the hypotheses regarding how chronic kidney disease (CKD) affects cognitive impairment is vascular damage in conjunction with malnutrition or inflammation. Moreover, compelling evidence demonstrates a decline in cerebral mean flow velocity and white matter hyperintensities with hemodialysis.

In connection with the above hypothesis, the factors that may potentially affect the cognitive abilities of hemodialysis patients are the condition of their blood vessels, as well as blood flow and oxygenation of their brain tissue. Arterial stiffness determined by pulse wave velocity (PWV) is one of the validated parameters that shows the overall condition of blood vessels in the body. Studies reveal that hemodialysis patients show increased vascular stiffness compared to patients with CKD stage 4 and patients after kidney transplantation [3]. Brain oxygenation can be non-invasively assessed using near-infrared spectroscopy (NIRS). It has been proven that hemodialysis patients show a significantly reduced rSO2 compared with the general population [4]. On the other hand, the most plausible hypothesis is that the damage may be caused by uremic (neuro) toxins produced in the course of CKD. It is also speculated that kidney failure prevents the production of neuroprotective factors, resulting in the suffering of the brain in CKD [5,6].

The association of a greater frequency of MCI beyond the age of 60 with the age-dominant group of dialysis patients—likewise 60 and older—is also reflected in the increased mortality, therefore, acquires therapeutic importance [7].

Cognitive impairment in hemodialysis patients is common and refers to many domains, such as cognitive-motor function, language, executive function, learning and memory, and complex attention. According to a study conducted, executive function and memory are the cognitive functions most closely linked to mortality [7,8].

Among cognitive function tests, the Montreal cognitive assessment test (MoCA) is characterised by the highest precision [9] with great sensitivity in the group of hemodialysis patients [10]. A recent Cochrane library review of the evidence also underlined the accuracy of the MoCA test for detecting dementia [11]. Computer-Based Cognitive Assessment Tool (CompBased-CAT)—CogniFit—is an advanced assessment made via a web browser or mobile app. It allows the assessment of specific cognitive abilities, such as concentration/attention, memory, reasoning, planning, or coordination. CompBased-CAT has already been validated in other groups of patients [12] and can be efficiently combined with an intervention tailored to the patient’s needs (training module).

The purpose of the study was to determine whether vascular stiffness, brain oxidation, comorbidities, and certain healthy behaviours impact cognitive impairment assessed by MoCA and CompBased-CAT in a cohort of hemodialysis patients.

## 2. Materials and Methods

Seventy-five hemodialysis maintenance patients were considered for the study at the academic dialysis centre. Study exclusion criteria included manual disability of the upper limbs, severe vision problems, being previously diagnosed and treated by a psychiatrist due to dementia or Alzheimer’s disease, post-stroke condition, lack of sign of informed consent, less than 3 months on renal replacement therapy, and the patient’s refusal to participate in the study. Patients have been adequately dialysed a minimum of 3 times a week and achieved a target Kt/V > 1.4. Each patient had the results of basic laboratory tests (urea, potassium, sodium, calcium, phosphates, parathyroid hormone, and haemoglobin level) taken before the dialysis session (Table 1). None of the patients was taking drugs influencing the central nervous system.

Finally, 20 men and 7 women completed all tests and measures in this pilot study (Figure 1). The average age is 51 years (21–80 years), and the average dialysis vintage is 2 years (Table 1). Patients were examined by trained personnel and completed a battery of cognitive tests: The MOCA test and, additionally, the multi-domain cognitive assessment battery by CogniFit™, which is a commercial online application and an example of Comp-Based-CAT. A cognitive function assessment was performed before the hemodialysis session. The Cognifit contains visual, auditory, and cross-modal tasks, including puzzles, problem-solving, and reaction time games. The CogniFit test was completed using a mobile application installed on the tablet (Galaxy Tab A6, Samsung electronics, Korea). All patients had been able to operate mobile devices (tablets, smartphones) before being tested. Due to the need to use both hands for some tasks, it would be impossible for patients with a dialysis fistula to complete the test during a dialysis session. Oxygen saturation (rSO2) of frontal cerebral lobes (INVOS 5100c system) and PWV (IEM Mo-bil-O-Graph) were measured. The INVOS 5100c system uses near-infrared spectroscopy to assess brain oxygenation non-invasively. The IEM Mo-bil-O-Graph uses oscillometric methods by detecting data from the cuff during inflation and converting it using patented algorithms to estimate PWV. Clinical and laboratory data were also recorded. Patient regular passive or active (reading, crossword solving, electronic games) behaviour during sessions was noted. For each patient, the Charlson Comorbidity Index (CCI) was calculated, which is a validated tool for assessing 10-year mortality from patient morbidity data. [13,14]. Physical activity levels were measured using the Physical Activity Rapid Assessment (RAPA), a self-administered questionnaire consisting of nine binary questions (answer yes or no) presented textually and graphically. The questionnaire had already been used in a group of elderly and hemodialysis patients [15,16].

With the potential intradialytic hypotension and feeling of exhaustion at the end of the HD session taken into account, all cognitive tests, as well as RAPA and behavioural anamnesis, were taken in the first hour of the session.

Statistical power (sample size estimation) analysis was conducted, determining the minimum r = 0.51, at which the test power was 0.8 (assumptions n = 27, α = 0.05). Multivariate regression analysis was performed among parameters showing significance in univariate analysis (no more than 3 parameters in each model tested).

All procedures performed in this study were in accordance with the ethical standards of our institutional research committee and with the 1964 Helsinki declaration and its later amendments. Informed consent was obtained from all individual participants included in the study.

## 3. Results

Patients’ characteristics, as well as measured results, are displayed in Table 1. The median MoCA score is 28 (IQR 23;29). The CompBased-CAT total score is 321 (IQR 212; 371). Median saturation (rSO2) is more profoundly reduced in the left frontal lobe when compared with the right (53% vs. 59%).

### 3.1. Univariate Analysis

The MoCA results in univariate analysis correlate with rSO2 front R (r = 0.44, *p* = 0.02), rSO2 front L (r = 0.62, *p* = 0.001), PWV (r = −0.69, *p* = 0.0001), CCI (r = −0.59, *p* = 0.001), RAPA (r = 0.72, *p* = 0.0001) (Figure 2). Statistically significant correlations were found between the CompBased-CAT result and rSO2 front R (r = 0.49, *p* = 0.009), rSO2 front L (r = 0.65, *p* = 0.0001), PWV (r = −0.64, *p* = 0.0001), CCI (r = −0.58, *p* = 0.002), RAPA (r = 0.56, *p* = 0.002) (Figure 3). Both in the case of correlation with MoCA and Cognifit rSO2, the R front did not reach the required test power. Additionally, the CompBased-CAT score correlates with MoCA (r = 0.85, *p* = 0.0001). The interrelationships between the parameters are presented in the correlation matrix (Figure 4).

In addition, patients who actively spend time on dialysis score higher in the CompBased-CAT and MOCA tests (CompBased-CAT: 386 vs. 233, *p* = 0.0002; MOCA: 28.6 vs. 24.7, *p* = 0.0002; and active vs. non-active, respectively) (Figure 5) and use social media (CompBased-CAT: 352 vs. 255, *p* = 0.03; MOCA: 27.8 vs. 25.2, *p* = 0.026; and users vs. non-media users, respectively) (Figure 6). Markedly higher scores in the CompBased-CAT test are obtained by non-smokers (370 vs. 250, *p* = 0.006, non-smokers vs. smokers, respectively); however, in the case of the MoCA test, the difference was not statistically significant (27.7 vs. 25.7, *p* = 0.09) (Figure 7). There was no correlation between the results of cognitive tests and the concentration of urea, potassium, sodium, phosphates, calcium, and haemoglobin before dialysis. In addition, smokers also showed increased vascular stiffness (mean PWV 8.9 vs. 6.9, *p* = 0.04) and less physical activity correlated with increased PWV (r = −0.47, *p* = 0.014).

### 3.2. Multivariate Analysis

Results of the multiple linear regression indicate that there is a very strong collective significant effect between the PWV, RAPA, and MOCA (F = 25.76, *p* < 0.001, R^2^ = 0.68, R^2^adj = 0.66) (Table 2). Multivariate analysis of the same parameters with the CompBased-CAT confirms their correlation with cognitive functions (F = 12.03, *p* < 0.001, R^2^ = 0.5, R^2^adj = 0.46). (Table 3). In addition, the correlation between rSO2 front L, RAPA and cognitive function is demonstrated—MOCA (F = 21.63, *p* < 0.001, R^2^ = 0.64, R^2^adj = 0.61) and CompBased-CAT (F = 13.15, *p* < 0.001, R^2^ = 0.52, R^2^adj = 0.48) (Table 4 and Table 5). In the multiple linear regression models, the power of the test was above 0.8. No other correlations were found using the multivariate model.

## 4. Discussion

In the current study, cognitive function (executive functions, in particular) measured by the MoCA test and Computer-Based Cognitive Assessment Tool was analysed. Moreover, their relationship to arterial stiffness (a surrogate for vessel damages), frontal lobes oxygen saturation, and healthy habits were analysed in a cohort of hemodialysis patients.

Cognitive impairment in hemodialysis patients, called since the 1960s “dialysis dementia”, is still a serious problem influencing patient compliance and, what is more relevant, also survival. In the COGNITIVE-HD study, CI occurred in 474/676 patients. It also occurs significantly more often in dialysis (36%) than in non-dialysed (25%) patients in Japan [17]. Additionally, hemodialysis is associated with a higher risk of CI than peritoneal dialysis, and renal transplantation significantly reduces CI symptoms [18,19]. In the last decades, many risk factors for the loss of cognitive abilities have been identified in hemodialysis patients. There are traditional factors, such as the level of education or the presence of depression and factors related to dialysis, e.g., dialysis vintage and the presence of specific inflammatory factors [20]. Awareness of these factors can help identify the patients most at risk of developing cognitive deficits. To assess cognitive function in hemodialysis patients, both the standardised Montreal cognitive assessment (MoCA) [21] and the novel CompBased-CAT are also good options for those above 60 years of age [12,22]. Such an approach seems to be a feasible assessment strategy for multimorbid older adults with or without cognitive impairment.

In this study, various risk factors of cognitive decline have been assessed. Parameters such as rSO2 of the frontal lobes, PWV, CCI score, physical activity, and the way of spending time during dialysis were examined. The first negative correlation was found between the result of cognitive tests and CCI (CompBased-CAT r = −0.57, MoCA r = −0.59). CCI has not yet been associated with cognitive impairment in the group of hemodialysis patients. Such a relationship has already been found among patients with mild-to-moderate Alzheimer’s disease [23]. Such a relationship has not been confirmed in a group of elderly people with dementia [24]. CCI may be another useful indicator of the risk of cognitive impairment in hemodialysis patients.

We measured the frontal lobe oxygen saturation (rSO2) to confirm the metabolic risk factor of dementia, which is more prominent in dialysis patients. It is mainly due to repeated brain hypoperfusion during hemodialysis sessions, namely, intradialytic blood pressure changes cause declines in cerebral oxygenation saturation during HD. It was recently confirmed in a cerebrovascular reactivity study using a combination of functional MRI and cerebral oxygenation saturation [25]. Studies show that cerebral flow measured using transcranial Doppler ultrasound to measure cerebral arterial mean flow velocity (MFV) is reduced in hemodialysis patients. [26]. The right frontal lobe is related to the formation of new cognitive processes, while the left frontal lobe is crucial “for the cognitive selection driven by the content of working memory and for context-dependent behaviour” [27]. The relationship between the frontal lobes and the results of cognitive tests (MoCA), mainly of executive function, has already been described [28]. In our study, the results of cognitive tests (MoCA, CompBased-CAT) also positively correlated with the saturation of both the left and right frontal lobes. The study clearly shows a stronger relationship between the left frontal lobe saturation and the result of cognitive tests. Further analysis using multifactorial models showed a correlation between RAPA, rSO2 front L, and cognitive functions. This may indicate that physical activity has a positive effect on blood flow through the left frontal lobe, which leads to better results in cognitive functions.

Multivariate regression analysis in our study indicated an independent impact of physical activity score (RAPA), as well as arterial stiffness (PWV), on cognitive function (MoCa and CompBased-CAT). The relationship between cognitive functions and vascular stiffness in a group of hemodialysis patients has already been described [29]. The relationship between physical exercise and cognitive ability is well-known among the general population. Increasing physical exercise may prevent cognitive impairment from developing [30], even in the elderly [31]. It also refers to chronic hemodialysis patients. Authors of a recent systematic review found that physical exercise might improve or at least not worsen cognitive performance in HD patients, whereas the effect of cognitive training has not yet been adequately studied [32]. There is a general feeling that we need more sensitive and specific cognitive tests to measure the effects of interventions in the HD population adequately. This is why we supplemented MoCA with CompBased-CAT, keeping the generational change in mind, which is also taking place among dialysis centre patients—more and more people will be willing to use mobile solutions. A strong correlation with the standardised MoCA questionnaire was obtained.

The next lifestyle habit negatively affecting cognitive function is smoking cigarettes [33]. In a study looking at brain perfusion in patients with ESRD on HD maintenance who also had cognitive impairment, a high number of cortical defects (frontal and temporal lobes) consistent with the multiple-infarct type of dementia were reported. The majority of the patients in the study were current or former smokers [34]. Both smoking and physical activity are modifiable risk factors. This opens the field for conducting interventional studies and for the patient’s work to reduce the risk of cognitive dysfunction.

This study also indicates that not only physical but also mental activity is very important in the context of cognitive decline. Patients who use social media and actively spend time during dialysis (computer games, crosswords vs. sleeping, watching TV) also obtained statistically significantly higher results in tests of cognitive functions. This hypothesis is made more likely by the fact that the longest-treated dialysis patients in the world show high mental activity, for example, Helena Garvao, who lived with 44 years of hemodialysis. The patient earned her PhD in linguistics and, at the age of 47, additionally completed studies in Portuguese literature. Furthermore, she worked as a lecturer for 30 years [35].

The cross-sectional design, the low recruitment rate and the relatively small sample size are the main limitations of the study. The potential selection bias of the study was that all patients must have used electronic tablets (some elderly persons might have refused to do it).

In summary, we confirmed that cognitive impairment in hemodialysis patients is multifactorial and healthy habits from the pre-dialysis time play a significant role. We urgently need diagnostic and preventive/therapeutic means in the cognitive field for this population. One must remember that, besides intradialytic interventions (reducing hypoperfusion by limiting ultrafiltration, modifying time, HD to HDF switch, etc.), also promoting physical/mental activity may change the risk of dementia progression. Further studies in this field are expedient.

## 5. Conclusions

Healthy habits, such as being physically active, not smoking, and inter-dialysis sessions (tasks and mind games, use of social media) are associated with better cognitive functions. Cognitive functions in hemodialysis patients are related to vascular stiffness (PWV), physical activity (RAPA), the blood supply to the frontal lobes (rSO2), and comorbidity (CCI).

## Figures and Tables

**Figure 1 jcm-12-02042-f001:**
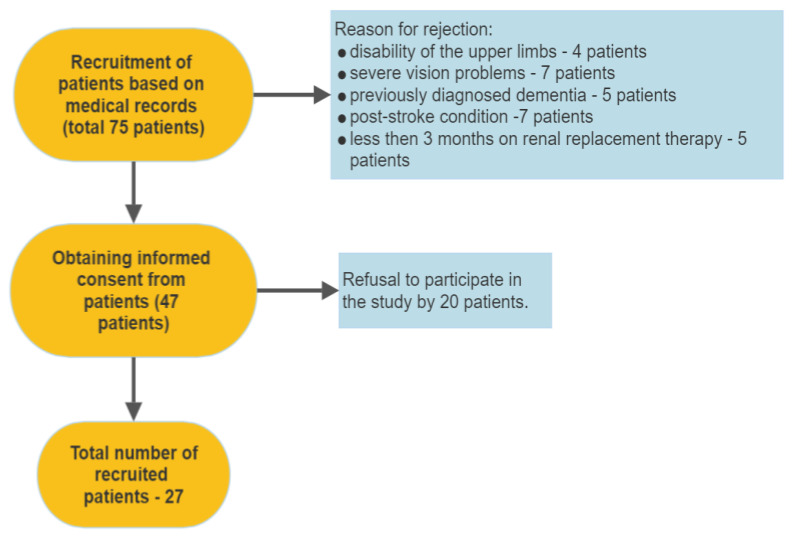
Recruitment process diagram.

**Figure 2 jcm-12-02042-f002:**
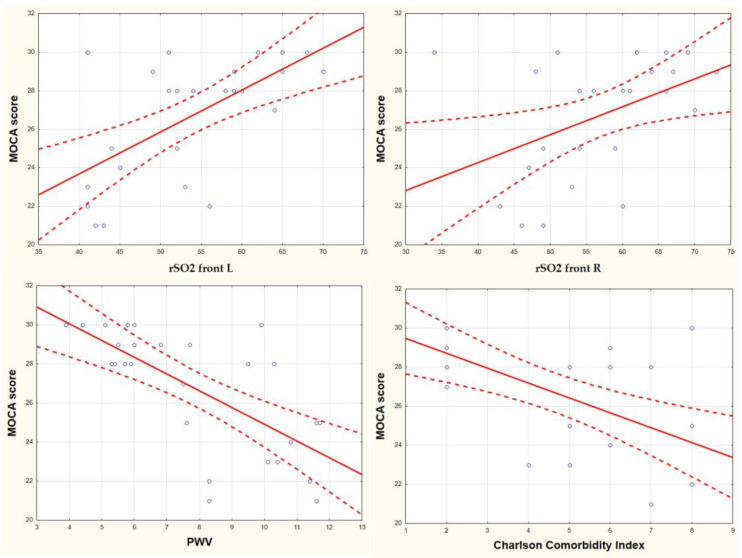
Correlation of the MOCA score with rSO2 front R, rSO2 front L, PWV, and CCI.

**Figure 3 jcm-12-02042-f003:**
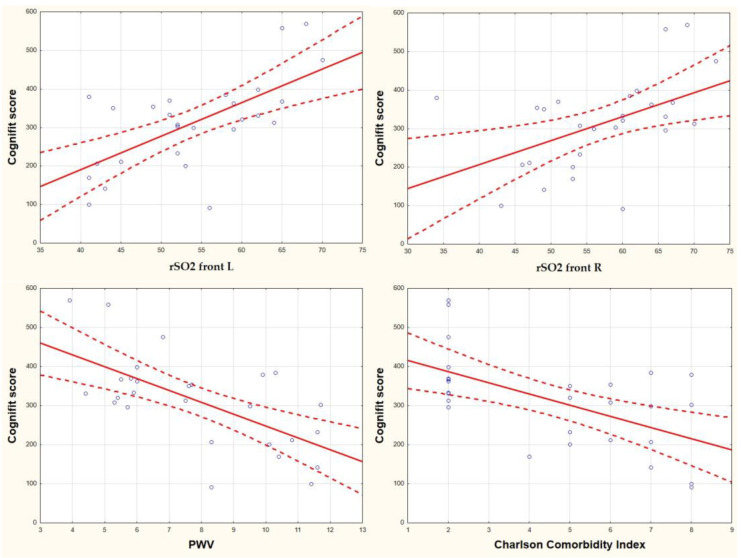
Correlation of the CompBased-CAT (Cognifit score) with rSO2 front R, rSO2 front L, PWV, and CCI.

**Figure 4 jcm-12-02042-f004:**
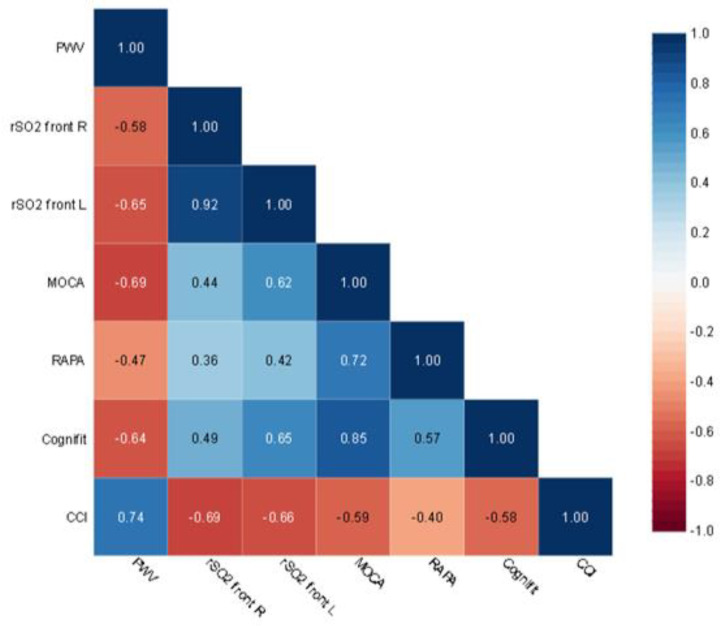
Correlation Matrix.

**Figure 5 jcm-12-02042-f005:**
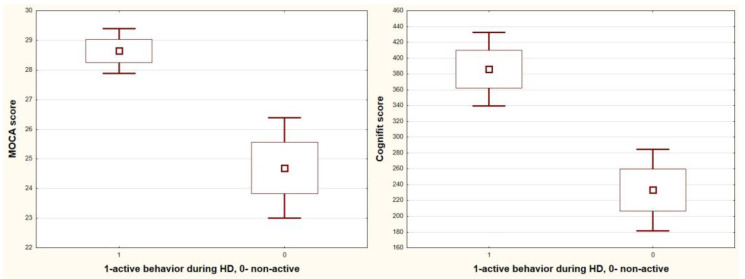
Box-whisker graph of CompBased-CAT (Cognifit score) and MOCA vs. HD session activity.

**Figure 6 jcm-12-02042-f006:**
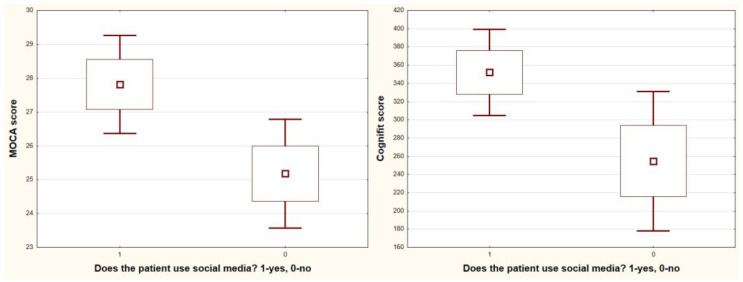
Box-whisker graph of CompBased-CAT (Cognifit score) and MOCA vs. using social media.

**Figure 7 jcm-12-02042-f007:**
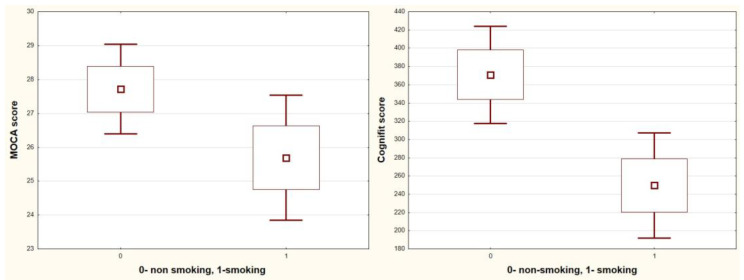
Box-whisker graph of CompBased-CAT (Cognifit score) and MOCA vs. smoking.

**Table 1 jcm-12-02042-t001:** Patients’ characteristics and results.

	Mean	Median (IQR)/%
Age	51.3	53 (34; 68)
BMI	25.5	24.5 (21.5; 32)
Dialysis vintage	2.37	2 (0.5; 3)
Smoking	13	48%
Diabetes	6	22.20%
Hypertension	16	59.20%
Hemoglobin g/dL	10.7	10.2 (9.6; 11.1)
Urea mg/dL	130	134 (78; 175)
Potassium mmol/L	5.5	5.69 (3.6; 6.1)
Sodium mmol/L	138	140 (131; 145)
Calcium mg/dL	8.9	8.7 (7.3; 12.1)
Phosphorus mg/L	6.3	5.9 (3.1; 10.8)
Parathyroid hormone pg/mL	829	857 (30; 1547)
Residual diuresis (>500 mL)	12	44.40%
Charlson Comorbidity Index (CCI)	4.6	5 (2; 7)
rSO2 front R **	57%	59 (49; 56)
rSO2 front L	54%	53 (49; 66)
MoCA	25.7	28 (23; 29)
RAPA	2.8	3 (0; 6)
Cognifit total score	312.7	321 (212; 371)
Processing speed	269	232 (117; 382)
Shifting of attention	261	214 (58; 384)
Visual short-term memory	254	199 (27; 350)
Auditory short-term memory	315	337 (200; 403)
Working memory	285	256 (188; 423)
Naming	327	373 (98; 528)

BMI—body mass index. ** rSO2 front—oxygen saturation of frontal lobe. R—right. L—left. MoCA—Montreal Cognitive Assessment. RAPA—Physical Activity Rapid Assessment.

**Table 2 jcm-12-02042-t002:** MOCA/PWV/RAPA multiple linear regression (adjusted R^2^ = 0.66; MOCA = 28.892 − 0.563 PWV + 0.810 RAPA).

	Coeff.	SE	t-Stat	Stand Coeff.	*p*-Value
PWV	−0.56	0.16	−3.5	−0.46	0.002
RAPA	0.81	0.21	3.91	0.51	0.001

**Table 3 jcm-12-02042-t003:** CompBased-CAT/PWV/RAPA multiple linear regression (adjusted R^2^ = 0.46; Cognifit = 433.605623 + 20.873369 RAPA − 22.85519 PWV).

	Coeff.	SE	t-Stat	Stand Coeff.	*p*-Value
PWV	−22.86	7.75	−2.95	−0.48	0.007
RAPA	20.87	9.99	2.09	0.34	0.047

**Table 4 jcm-12-02042-t004:** MOCA/rSO2 front L/RAPA multiple linear regression (adjusted R^2^ = 0.61; MOCA = 16.918 + 0.889 RAPA + 0.135 rSO2 front L).

	Coeff.	SE	t-Stat	Stand Coeff.	*p*-Value
rSO2 L	0.14	0.05	2.88	0.39	0.008
RAPA	0.89	0.21	4.15	0.56	0.0003

**Table 5 jcm-12-02042-t005:** CompBased-CAT/rSO2 front L/RAPA multiple linear regression (adjusted R^2^ = 0.48; Cognifit = −109.901 + 21.790 RAPA + 6.681 rSO2 front L).

	Coeff.	SE	t-Stat	Stand Coeff.	*p*-Value
rSO2 L	6.68	2.09	3.2	0.5	0.004
RAPA	21.79	9.52	2.29	0.36	0.031

## Data Availability

Not applicable.

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
