# Peer review of "The Influence of Healthy Habits on Cognitive Functions in a Group of Hemodialysis Patients"

_jcm, 2023, doi:10.3390/jcm12052042_

Round 1

Reviewer 1 Report

Comments to the authors

1.                   The methods are described briefly. For example, the study used PWV measurements. This method needs to be described.

2.                   Table 1: the study cohort is not adequately described – please add laboratory parameters, dialysis-related parameters, medications.

3.                   Results: the study reports uni-variate correlations between the MOCA score and PWV as well as between MOCA score and rSO2. The authors should provide multi-variate adjusted correlations.

4.                   Results: the study provides some sub-group comparisons that are not clear. The sample size of the study is so small and these analyses are exploratory and have low statistical power.

5.                   The overall recruitment rate in this study is very low. Figure 1 indicates that 20 patients refused to undertake the protocol procedures. Please explain the reasons. What measures were taken to increase the sample size of your study?

6.                   The limitations of the study, such as its cross-sectional design, the low recruitment rate, the small sample size, need to be acknowledged in the discussion.

Author Response

Dear Reviewers and Editor of JCM,

We are gratefully for deep insight and suggestions how to improve  the manuscript. Please accept our responses and explanations given below.  Thank you so much for your time and expertise.

Reviewer #1

  1. The methods are described briefly. For example, the study used PWV measurements. This method needs to be described.

Answer: Thank you for this remark.  The methods used including PWV (IEM, Mobil-O-Graph) has been broader described in “material and methods” in revision of the manuscript.

  1.                   Table 1: the study cohort is not adequately described – please add laboratory parameters, dialysis-related parameters, medications.

Answer: Following this important suggestion we added available laboratory parameters, especially reflecting dialysis adequacy in table 1. Non of patients was taking drugs acting on CNS or due to dementia.

  1.                   Results: the study reports uni-variate correlations between the MOCA score and PWV as well as between MOCA score and rSO2. The authors should provide multi-variate adjusted correlations.

Answer: Thank you for this critical remark. Report from multivariate analysis has been performed and results displayed in tables 2-5. Consequently finding from this regression analysis are taken into account in discussion.

5.                   The overall recruitment rate in this study is very low. Figure 1 indicates that 20 patients refused to undertake the protocol procedures. Please explain the reasons. What measures were taken to increase the sample size of your study?

Answer: We agree that recruitment rate was low and refusals high. One of reason is COVID-19 pandemic restrictions. The imposed restrictions on interpersonal contacts sometimes raised exaggerated fears. The measures we tried for increasing sample size were a two-level way of talking, explaining the purpose of the study (PhD student --> professor) and demonstrating a potential training option as a new way of training the mind.

Despite the small study group, statistically significant results and, in most cases, sufficient test power were achieved. Statistical power (sample size estimation) analysis was conducted determining the minimum r=0.51, at which the test power (TP) is 0.8 (assumptions n=27, α=0.05).  One of the problems of recruiting patients was to convince them to use mobile equipment. The largest group in the clinic dialysis center are elderly patients (the average age in the entire dialysis center was 64 years), who rarely use tablets and smartphones on a daily basis.

  1.                   The limitations of the study, such as its cross-sectional design, the low recruitment rate, the small sample size, need to be acknowledged in the discussion.

Answer: Your point is very important and appropriate sentence has been incorporated in  last part of discussion section.

All your  suggestions has been incorporated. Thank you for remarks.

Reviewer 2 Report

The authors investigated the cognitive function in hemodialysis patients. The topic of this study is interesting and critical in the hemodialysis field.

However, there are some concerns in this study.

Major points

               First, the aim of this study is quite unclear. Even though the authors described that they aimed to compare MoCA Test with Computer-Based cognitive assessment tool in the study, they only showed various correlations for rSO2, PWV, RAPA in the two tests. In other words, they did not show a direct comparison between the two methods. I understand that Computer-based cognitive assessment might be better for the cognitive assessment of non-smokers. However, I do not think that the Computer-based cognitive assessment is superior to MoCA test. The authors should revise the whole manuscript to support their hypothesis. The present manuscript will confuse readers. The abstract should be revised as well.

               Second, this study was conducted using an electric tablet, which could cause bias. More than 40% of eligible patients refused to participate in this study. This might be because they were not willing to use electric appliances. This could be a big selection bias. In addition, people who used social media should be good at using tablets and will get a higher score. The authors should state these limitations clearly in the discussion.

               Third, the authors should show when and where the study was conducted. It is very important to show the timing of this study. If this study was conducted after the dialysis, some patients might have been suffering from hypotension after the treatment.

Minor points

               There are numerous unmeaningful hyphenated words in this manuscript, such as “hy-potheses”, and “as-sessing”. The authors should correct them properly.

Author Response

Dear Reviewers and Editor of JCM,

We are gratefully for deep insight and suggestions how to improve  the manuscript. Please accept our responses and explanations given below.  Thank you so much for your time and expertise.

Reviewer #2

Major points

               First, the aim of this study is quite unclear. Even though the authors described that they aimed to compare MoCA Test with Computer-Based cognitive assessment tool in the study, they only showed various correlations for rSO2, PWV, RAPA in the two tests. In other words, they did not show a direct comparison between the two methods. I understand that Computer-based cognitive assessment might be better for the cognitive assessment of non-smokers. However, I do not think that the Computer-based cognitive assessment is superior to MoCA test. The authors should revise the whole manuscript to support their hypothesis. The present manuscript will confuse readers. The abstract should be revised as well.

Answer:
Thank you for this critical remark.  The purpose of the study was to determine which behavioral and clinical factors including vascular condition are associated in cognitive impairment in cohort of hemodialysis patients. To make it clear following changes has been made in the manuscript:

  1. We revised whole manuscript adding clear aim in the introduction and abstract. We removed unclear paragraphs/sentences or moved to different parts
  2. The abstract was rewritten. The results from multivariate regression analysis are added
  3. New references has been added 32 and 34.

In fact our aim was not to directly compare MoCA Test with Computer-Based cognitive assessment tool. We used it as additional tool inspired by paper by Bogataj et al. who pointed that we need more specific cognitive tests to adequately measure the effects of interventions in the HD population [Front Public Health. 2022 Oct 14;10:1032076. doi: 10.3389/fpubh.2022.1032076 ]. This is why we supplemented MoCA with CompBased-CAT keeping in mind generational change, which is also taking place among dialysis center patients. More and more people will be willing to use mobile solutions. In addition to the assessment itself, the applications also offer cognitive function training, which may reduce the risk of cognitive function loss. A strong correlation with the standardized MOCA questionnaire was obtained confirming its usefulness.

               Second, this study was conducted using an electric tablet, which could cause bias. More than 40% of eligible patients refused to participate in this study. This might be because they were not willing to use electric appliances. This could be a big selection bias. In addition, people who used social media should be good at using tablets and will get a higher score. The authors should state these limitations clearly in the discussion.

Answer: Thank you for this critical remark.   Following sentence has been added in the discussion section: “ Cross-sectional design, the low recruitment rate and relatively small sample size are main limitations of the study.  Potential selection bias of the study was that all patients must used electronic tablets (some elderly persons might refuse to use it).”

               Third, the authors should show when and where the study was conducted. It is very important to show the timing of this study. If this study was conducted after the dialysis, some patients might have been suffering from hypotension after the treatment.

Answer: Thank you for another excellent point. Taking into account factors like intradialytic hypotension and feeling exhausted at the end of HD session all cognitive tests were taken in the first hour of session. This information has been added in method section.

Minor points

               There are numerous unmeaningful hyphenated words in this manuscript, such as “hy-potheses”, and “as-sessing”. The authors should correct them properly.

Answer: Thank you for this remark. Appropriate corrections has been made.

All your  suggestions has been incorporated in the manuscript. Thank you very much for contributing to higher quality of the paper.

Round 2

Reviewer 1 Report

No further comments.

Author Response

Dear Reviewer,

We appreciate  your comments contributing to better quality of the paper

Reviewer 2 Report

The authors addressed my concerns.

However, there are still a few issue now.

As this study is conducted in a cross-sectional study, the description in the abstract “Inter-dialysis healthy behaviors, such as physical activity, not smoking, and intra-dialysis activities (tasks and mind games), are connected with improved cognitive abilities.” was overstated.

In addition, the word limit of the abstract is about 200 words in this journal. The authors should write the result and discussion objectively and concisely.

P values and other parameters in Table 2-5 should be rounded.  P=0.00183032 is not common in scientific literature.

There are several grammatical mistakes, such as “Potential selection bias of the study was that all patients must used electronic tablets (some elderly persons might refuse to use it).”

Author Response

Dear Reviewer,

We appreciate  your comments contributing to better quality of the paper.

In response to raised points we have made following improvements:

As this study is conducted in a cross-sectional study, the description in the abstract "Inter-dialysis healthy behaviors, such as physical activity, not smoking, and intra-dialysis activities (tasks and mind games), are connected with improved cognitive abilities." was overstated.

Response: Thank you for this remark. The abstract has been corrected and shortened according to limitation of 2000words.  

In addition, the word limit of the abstract is about 200 words in this journal. The authors should write the result and discussion objectively and concisely

Response: please see response to previous point.

P values and other parameters in Table 2-5 should be rounded.  P=0.00183032 is not common in scientific literature.

Response: That was our mistake which is corrected in revision of manuscript

There are several grammatical mistakes, such as "Potential selection bias of the study was that all patients must used electronic tablets (some elderly persons might refuse to use it).

Response: The whole manuscript underwent careful editing by native speaker and checked by software for grammatical correction. All changes in manuscript are marked in green.

We are very thankful for all remarks and suggestions regarding our manuscript.